# siRNA Features—Automated Machine Learning of 3D Molecular Fingerprints and Structures for Therapeutic Off-Target Data

**DOI:** 10.3390/ijms26146795

**Published:** 2025-07-16

**Authors:** Michael Richter, Alem Admasu

**Affiliations:** 1Department of Chemistry, Binghamton University, Binghamton, NY 13902, USA; 2Department of Physics and Astronomy, Rutgers University, Piscataway, NJ 08854, USA; asa123@rutgers.edu

**Keywords:** siRNA, off-target prediction, chemical modification, argonaute, hAgo2, structure-based modeling, extended connectivity fingerprints (ECFPs), structural modeling, machine learning, feature engineering, molecular dynamics

## Abstract

Chemical modifications are the standard for small interfering RNAs (siRNAs) in therapeutic applications, but predicting their off-target effects remains a significant challenge. Current approaches often rely on sequence-based encodings, which fail to fully capture the structural and protein–RNA interaction details critical for off-target prediction. In this study, we developed a framework to generate reproducible structure-based chemical features, incorporating both molecular fingerprints and computationally derived siRNA–hAgo2 complex structures. Using an RNA-Seq off-target study, we generated over 30,000 siRNA–gene data points and systematically compared nine distinct types of feature representation strategies. Among the datasets, the highest predictive performance was achieved by Dataset 3, which used extended connectivity fingerprints (ECFPs) to encode siRNA and mRNA features. An energy-minimized dataset (7R), representing siRNA–hAgo2 structural alignments, was the second-best performer, underscoring the value of incorporating reproducible structural information into feature engineering. Our findings demonstrate that combining detailed structural representations with sequence-based features enables the generation of robust, reproducible chemical features for machine learning models, offering a promising path forward for off-target prediction and siRNA therapeutic design that can be seamlessly extended to include any modification, such as clinically relevant 2′-F or 2′-OMe.

## 1. Introduction

Despite the growing interest in leveraging machine learning and computational techniques for chemical data and siRNA therapeutics, a comprehensive exploration of feature engineering in this domain remains significantly underdeveloped. A thorough review of the recent literature (2023–present) using advanced search strategies in Web of Science (WOS) yielded only a single relevant review article [1]. This scarcity of resources highlights the novelty and unexplored nature of applying feature engineering and machine learning to chemically modified siRNA design and off-target prediction. The current gap underscores the need for new studies to establish foundational frameworks and methodologies in this critical intersection of cheminformatics, RNA therapeutics, and machine learning.

Small interfering RNAs (siRNAs) are short, double-stranded RNA molecules that use the RNA interference (RNAi) pathway to silence gene expression post-transcription. By guiding the RNA-induced silencing complex (RISC) to complementary mRNAs, siRNAs induce targeted degradation, effectively preventing the production of specific proteins. Owing to their precision and broad applicability, siRNAs hold promise for treating a variety of conditions, including cancer, genetic disorders, and viral infections.

A major challenge in siRNA therapeutics is the unintended off-target effects, where siRNAs bind to partially complementary mRNA sequences or interact with unintended proteins. These off-target interactions can lead to undesirable gene silencing, toxicity, and immune activation, undermining their therapeutic efficacy and safety. The seed region of the siRNA guide strand (positions 2–8) is particularly prone to off-target binding, as it drives initial mRNA recognition. To address these issues, chemical modifications have emerged as powerful tools to enhance siRNA specificity and reduce off-target interactions, and are now considered the standard in modern siRNA therapeutic design.

The discovery of effective chemical modifications for siRNA faces several challenges. These include balancing stability, efficacy, and safety, and especially minimizing off-target effects. Chemical modifications must preserve RNAi activity while reducing off-target interactions. However, understanding how specific modifications impact the complex interplay of siRNA–protein and siRNA–mRNA interactions remains limited. Advances in computational modeling and high-throughput experimental platforms provide new opportunities to explore novel chemical structures and predict their biological outcomes. Nevertheless, because these chemical modifications often involve noncanonical or “unnatural” structural elements, conventional sequence-based off-target prediction methods cannot adequately capture them, thereby reinforcing the need for new approaches. Integrating these approaches is critical for optimizing siRNA therapeutics to address unmet clinical needs [2].

As this is an emerging field, a review of the recent literature was conducted to capture the latest innovations in siRNA off-target prediction, feature engineering, and structure-based modeling prior to formal peer-reviewed publication. Several significant works have appeared within the last year, covering a range of topics, including machine learning approaches, molecular modeling, bioinformatics tools, and structural biology insights.

Oshunyinka introduced a machine-learning framework using sequence-derived features to predict siRNA potency [3]. His model, tested against SARS-CoV-2, underscored how the careful encoding of siRNA sequences can capture structure–potency relationships. Similarly, Bai et al. presented OligoFormer, a transformer-based deep learning model that jointly predicts siRNA knockdown efficacy and off-target potential [4]. The model incorporates thermodynamic parameters and pretrained RNA embeddings, and Bai et al. further validated it using microarray data and integrated TargetScan and PITA to refine off-target assessments. Long et al. proposed a GNN-based approach (published as siRNADiscovery) that represents siRNA–target pairs as a bipartite graph, leveraging message-passing to capture long-range dependencies in a sequence context. While that study focused on efficacy, the authors noted that it could potentially be extended to model off-target interactions [5].

Recent advances in feature engineering underscore the importance of chemical and sequence-based featurization. In a pair of complementary approaches, Liu et al. introduced Cm-siRPred—a machine learning model that integrates MACC-based molecular fingerprints, k-mer encoding, and target site accessibility metrics for chemically modified siRNAs and AttSiOff, a self-attention-based model coupling efficacy prediction with an off-target filtration module that ranks 3′ UTR sequences using seed matches [6,7]. Yadav et al. developed a stochastic model of siRNA endosomal escape, showing how incomplete cytosolic release can limit effective siRNA concentration and potentially increase off-target effects at high doses [8]. Cazares et al. released SeedMatchR, an R package that annotates RNA-seq data from knockdown experiments to flag differentially expressed genes harboring 6–8 mer seed matches, providing an open-source workflow for off-target analysis [9].

Beyond machine learning, several structural biology preprints have yielded important insights into siRNA–hAgo2 interactions. Sarkar et al. reported a 3.16 Å cryo-EM structure of hAgo2 bound to a guide siRNA and target mRNA, revealing a distortion at position 6 that repositions Lysine-709 in the catalytic site—a previously unobserved mechanism [10]. Wallmann and Van de Pette conducted a comparative evolutionary analysis of Argonaute, identifying conserved atomic interactions that stabilize the siRNA guide strand and suggesting that future models incorporate protein-contact features [11]. Finally, Enoki et al. combined explainable AI with wet-lab validation to discover a novel RNA-binding protein that modulates selective siRNA loading into RISC complexes, highlighting AI-guided bio-discovery’s potential [12].

Collectively, this research reinforces the pivotal role of structural insights, feature engineering, and data-driven models in refining siRNA design, emphasizing the growing reliance on sophisticated machine learning approaches to tackle off-target complexities. Machine learning (ML) has become indispensable for siRNA efficacy and off-target prediction, as it can uncover complex patterns beyond rule-based design. However, the quality of input features is critical to model success [13]. Traditional sequence encodings (e.g., one-hot vectors or positional indices) and simple chemical descriptors (e.g., SMILES strings) often fall short, as they primarily capture linear information and overlook the spatial or interaction context crucial for RNA–protein complexes [13,14]. Recent studies have explored advanced feature engineering to address this gap. For instance, graph neural networks (GNNs) show promise in small-molecule tasks but face scalability challenges with large biomolecular assemblies like siRNA–hAgo2 complexes [15]. Current GNN frameworks (e.g., DGL) are not optimized for the size and structural complexity of protein–RNA interactions, limiting their effectiveness for siRNA off-target modeling. This has motivated the development of alternative feature representations that can capture the three-dimensional and chemical nuances of siRNA molecules.

Researchers are increasingly combining sequence-based features with structure-based and physicochemical features. Platforms like iFeatureOmega integrate diverse encodings—from k-mer frequencies and position-specific scoring matrices to biochemical property profiles—yielding richer representations of RNA sequences [13]. Such integrative features can improve model performance but also risk overfitting, especially with limited training data. Therefore, a balance between feature richness and generalizability is essential.

Cheminformatics tools offer powerful solutions to the presentation of chemical modifications in siRNAs. Extended-connectivity fingerprints (ECFPs), for example, encode molecular substructures into fixed-length vectors [16]. Unlike NMR-derived fingerprints, which require physical compounds [17], ECFPs can be computed in silico to capture the local atomic neighborhoods of modified nucleotides. In our work, we employ ECFPs–to our knowledge, a novel application in the siRNA off-target context—which effectively encode both standard and chemically modified nucleotides. The early results indicated that compressed ECFP features achieved a performance comparable to explicit SMILES embeddings [18]. This demonstrates ECFP’s potential as a robust, information-rich descriptor of siRNA chemical diversity.

Innovative featurization approaches continue to emerge. One method converts 3D biomolecular structures into 2D images for convolutional neural networks (CNNs) [19]. By projecting protein 3D coordinates (or RNA–protein complexes) onto image channels, CNNs can learn spatial features relevant to binding interactions—essentially treating structure prediction as an image recognition problem. Another strategy focuses on thermodynamic stability features: calculating melting temperatures for different regions of the siRNA duplex. Kobayashi et al. showed that the stabilities of the seed region (positions 2–8) versus the 3′ non-seed region (e.g., positions 9–14) have opposite correlations with off-target silencing [20]. Incorporating such features, including interaction terms combining seed and non-seed stability, provides a biologically interpretable way to enhance off-target prediction.

All these efforts underscore a common theme: spatial- and interaction-specific features are key to improving ML models for siRNA. Conventional encodings that ignore the 3D context—a limitation noted in analogous domains like chromatin interaction prediction [14]—can miss critical determinants of binding affinity and specificity. By integrating structural data (e.g., distances in an siRNA–hAgo2 complex) and interaction-aware features (e.g., seed pairing stability), we aim to bridge this gap and boost model generalization.

Argonaute proteins are at the core of the RNAi pathway, binding siRNAs and guiding them to complementary mRNAs for cleavage. The guide strand’s 5′ end docks into a hydrophilic pocket formed at the interface of the MID and PIWI domains, anchoring the RNA and positioning it for target recognition. This structural organization ensures precise positioning of the siRNA guide for accurate gene silencing. However, only guide nucleotides g2–g4 are fully exposed and available for initiating interactions with target RNAs, while the remaining guide nucleotides are initially shielded. This pre-organized seed region facilitates rapid and efficient target recognition by lowering the thermodynamic cost of hybridization. Furthermore, the central cleft of Argonaute accommodates the siRNA in an extended conformation, with almost all interactions with the protein mediated through the RNA sugar-phosphate backbone [21]. While this is true, every atom of the siRNA—from the nucleobases to the backbone—contributes to its overall structural stability, dynamic behavior, and ability to mediate precise interactions. This underscores the importance of incorporating atomic-level detail when modeling siRNA–hAgo2 interactions to fully understand their functional impact.

Molecular modeling, especially energy minimization, offers a realistic view of siRNA–hAgo2 interactions through detailing the spatial arrangements and energetic considerations. Translating these characteristics into ML features can enhance predictive power, potentially improving both efficacy and safety in siRNA therapies. This approach parallels the advancements in CRISPR–Cas9 research, where integrating sequence-based features (e.g., nucleotide motifs and positional importance) with non-sequence data (e.g., structural properties, chromatin accessibility, and target site epigenetics) has significantly improved gRNA activity predictions [22].

By adopting a similarly holistic strategy, combining spatial, energetic, and sequence-based features, siRNA predictive models can achieve greater biological relevance and precision. We hypothesize that the structural features within hAgo2, including guide strand docking, seed region exposure, and interactions mediated by the sugar–phosphate backbone, all contribute to the specificity and efficiency of RNAi. Capturing these features at an atomic level and integrating them into machine learning models will enable more accurate predictions of siRNA efficacy and off-target effects, ultimately improving therapeutic design. We introduce a feature generation strategy that incorporates Argonaute interactions via structural modeling. This proof-of-concept explores whether modeling-derived features can match or exceed conventional approaches, thereby uniting structural biology and ML for more biologically relevant siRNA predictions. Building on current approaches, we developed a novel approach to enhance siRNA feature representation by incorporating 3D spatial information through molecular modeling and simulation (see Figure 1 for a schematic of this workflow). This methodology aimed to overcome conventional feature limitations by integrating chemically modified siRNA structures into predictive models through a modular workflow that can adapt to any modification, including 2′′-F or 2′′-OMe.

## 2. Results and Discussion

### 2.1. Initial Trials

Initial trials compared the performance of feature sets using a 1DCNN model implemented in TensorFlow.16 Among the tested datasets, Dataset 0 achieved the highest average precision score. While all trained datasets exhibited similar performance ranges, Dataset 0 consistently outperformed the others. However, the differences in precision across the datasets were modest, suggesting comparable efficacy among the initial feature sets.

Next, we optimized the criteria and evaluated the true positives, false positives, and accuracy for Dataset 0. Unfortunately, higher true positive counts (TP > 500) corresponded to lower accuracy (0.4562–0.5667), whereas low true positives (TP ≤ 10) yielded higher accuracy (0.7106–0.7112). This suggests a model bias toward zero outputs, inflating accuracy when true positives are underpredicted. Refinements are needed to balance accuracy and true positive counts. Using Dataset 4, precision improved up to 0.50, and accuracy reached 73%, showing a better performance in terms of traditional metrics.

### 2.2. Autogluon

To further enhance model performance and optimize feature utilization, we employed AutoGluon, a robust automated machine learning framework [23]. AutoGluon simplifies the process of identifying optimal models by leveraging its leaderboard function to rank model candidates based on their performance. This enabled the efficient selection of the best model for each dataset, followed by customized training. AutoGluon also excels at handling diverse data types, automatically tuning hyperparameters, and prioritizing various metrics for improved interpretability and predictive power. Table 1 summarizes the datasets that were used for training (described in detail below in 3.2 Feature Description). Across datasets (excluding Datasets 0 and 1), true positives (TP) ranged from 304 to 497, false positives (FP) from 175 to 267, precision from 0.61 to 0.67, accuracy from 74% to 78%, and TP/FP ratios from 1.59 to 2.01. Dataset 6R with LightGBM achieved the best overall TP/FP ratio of 2.

To rigorously evaluate feature sets and optimize predictive performance beyond initial trials, we employed AutoGluon [23], a framework for automated machine learning (AutoML). AutoGluon streamlines model selection, hyperparameter tuning, and ensembling. It is important to note that while Area Under the Precision–Recall Curve (AUPRC) was ultimately prioritized for final model evaluation due to the inherent class imbalance (see Appendix A), AutoGluon utilized different internal metrics (e.g., accuracy for HPO, roc_auc_ovo_macro for stack level optimization) that are appropriate for guiding specific optimization phases, as detailed in Appendix A. Initial model performance, assessed using AutoGluon’s default settings and leaderboard function before extensive optimization, is summarized in Table 2, with equations provided in Appendix A.

### 2.3. Feature Importance Analysis

To evaluate the critical features driving siRNA efficacy and off-target prediction, feature importance scores were calculated across all datasets using machine learning models. These scores highlight the most influential properties, including chemical fingerprints, residue-to-residue distances, and protein regions critical for siRNA–hAgo2 interactions. Figure 2 summarizes the results, showcasing the relative importance of features across nine datasets. The analysis underscores key trends in how structural and contextual features impact model predictions and provides valuable insights for refining feature generation.

Dataset 2 provides a numeric representation of the siRNA/target duplex, while dataset 3 encodes chemical fingerprints of the same information. In both datasets, the feature importance algorithm identified the ends of the target sequence as having the greatest influence on model predictions. To simplify visualization, features identified as important were relabeled as either 5′ target or 3′ target to indicate their specific direction relative to the target mRNA. These ends tend to overhang after modeling, exhibiting greater structural flexibility compared to the central regions, which will be stabilized through several interactions. Duplexes with higher instability (from mismatches) are likely to experience even greater fluctuations at the terminal positions, and the model was likely able to correlate these regions to the experimental activity levels.

Dataset 4 revealed that the last two gene index values were highly significant due to the sequential assignment of Ensembl identifiers, which inherently grouped related genes. Encoding these identifiers as three uint8 values allowed the higher-order bytes to capture numerical similarities, effectively clustering genes with similar sequences. This explains their strong influence during machine learning training. This finding demonstrates how a simple encoding strategy can inherently group related genes (Table 3). The high importance of *gene_idx* features highlights an interesting artifact: the sequential nature of Ensembl identifiers implicitly encodes gene family or functional relationships. By splitting the identifier into uint8 components, the model effectively learned a form of latent gene clustering without the use of explicit biological pathway information, demonstrating the power and potential pitfalls of using such indices as features.

Datasets 6 and 7 included structural features derived from modeling and minimizations. The results highlighted protein residues 790 and 791, which can be found in the interaction map shown in Figure 3, denoted by the light green color on the bottom left corner. Notably, Y790 interacts with backbone phosphates at positions 3–4 as well as 4–5 of the guide siRNA. Y790 is well known as an important contact point, but the role of residue V791 remains unclear and is not visible in this interaction diagram. V791, located adjacent to Y790 and positioned directly below Y790 on this map, may undergo conformational changes during minimization. The identification of these residues as critical determinants aligns with structural knowledge of hAgo2; these residues form part of the binding pocket interacting with the siRNA backbone near the modification sites [21,24]. This convergence between data-driven feature importance and known structure–function relationships validates the biological relevance of the computationally derived structural features.

### 2.4. Hyperparameter Optimization

To further enhance model performance, we utilized AutoGluon’s Hyperparameter Optimization (HPO) framework, which efficiently automates the process of tuning hyperparameters to achieve optimal configurations for machine learning models. By leveraging AutoGluon’s capabilities, the approach ensures a systematic exploration of the parameter space, streamlining model development and maximizing predictive performance across multiple metrics, including accuracy, precision, recall, specificity, and F1-score (Figure 4). The validation set and hyperparameter tuning configuration, as well as the training code used, are shown in Appendix A.

For datasets 0–4, accuracy scores consistently exceeded 0.7 across all models, with precision scores ranging from 0.65 to 0.8 for Datasets 2, 3, and 4. Specificity scores were notably high across the board, surpassing 0.85, indicating strong model performance in identifying true negatives. However, recall and F1-scores showed significant variability, with F1-scores only exceeding 0.5 when using KNN. Inflated specificity scores were observed in cases where true negatives (TNs) were disproportionately low, further complicating the interpretation of performance metrics.

With minimized data, accuracy and specificity remained high across all models, with precision reaching close to 0.9 for XGBoost (XGB) on Dataset 6R. However, this performance was misleading as XGB consistently failed to correctly classify the minority class, as reflected by the low recall and F1-scores. In contrast, KNN demonstrated the most balanced performance, achieving recall and F1-scores of approximately 0.5 while maintaining high accuracy and specificity (above 0.85). The precision scores for KNN were also satisfactory, exceeding 0.55, making it the most reliable model for minimized datasets.

These findings underscore the importance of balancing metrics during model evaluation. While certain models, such as XGBoost, excelled in terms of precision and specificity, their inability to address class imbalance limited their practical utility. Conversely, KNN provided a more consistent performance across metrics, offering a viable solution for off-target prediction in siRNA studies. Notably, across most metrics, Dataset 3 (positional ECFPs) consistently outperformed Dataset 4 (compressed ECFPs + gene index), as is visible in Figure 4. This suggests that the high-dimensional, chemically rich positional fingerprint representation captures more predictive information regarding off-target interactions than the combination of simplified modification fingerprints and the indirect gene clustering provided by the Ensembl ID features.

### 2.5. Metric Optimization

Metric optimization was performed to address the mixed results observed during hyperparameter optimization (HPO). By employing multi-model ensembles and testing 25 different metrics, the focus was placed on improving predictions for the minority class; the results are compiled in Figure 5 along with the complete confusion matarix data shown in Appendix A. This approach led to accuracy scores consistently approaching 0.8, though precision scores slightly declined, stabilizing above 0.7. Notably, recall and F1-scores improved significantly for Datasets 2, 3, and 4, which were based on simple encodings, chemical fingerprints, and hybrid feature strategies without structural modeling, reaching nearly 0.6.

For minimized datasets (6R and 7N), however, the recall and F1-scores were lower compared to non-minimized datasets, indicating a trade-off between model complexity and generalization. To illustrate this discrepancy, a black line was added to the figure, representing the HPO-optimized results for these datasets as a reference point. This comparison underscores the challenges of balancing performance across diverse datasets and optimization methods. Ultimately, metric optimization demonstrated its value in improving minority class predictions, particularly for non-modeled datasets, while emphasizing the need for dataset-specific approaches to enhance overall siRNA off-target prediction. The abbreviated training code used to search hyperparameters for LightGBM, XGBoost, and KNN is shown in Appendix A. Appendix A detail the cross-validation performance, as well as an assessment of random accuracy, as presented by [25].

### 2.6. Stack Level and Time Optimization

Following metric optimization, stack-level and time optimization was explored using AutoGluon’s ensemble framework, which integrates base models into meta-models at successive stack levels. Notably, despite representing the same underlying data (Figure 6), Dataset 3—employing detailed positional ECFPs for both siRNA and target mRNA (1345 features)—significantly outperformed Dataset 2, which used a simple numerical encoding for bases and modifications (43 features). Despite representing the same underlying biological interactions, the richer, higher-dimensional ECFP features of Dataset 3 showed superior discriminatory power for the machine learning models, enabling a more precise characterization of the chemical determinants of off-target effects and achieving AUPRC scores approaching 0.8 (peak AUPRC = 0.785, see Appendix A; Figure 6 shows AUPRC = 0.784).

Given that siRNA off-target prediction remains an undeveloped area in machine learning, with few direct comparisons, our model’s AUPRC scores of 0.784 and 0.736 are in contrast to the 0.714 reported for BERT-siRNA [26], suggesting differences in how feature encodings and optimization strategies influence precision–recall tradeoffs. While AUPRC is the optimal metric for siRNA off-target prediction, Spearman correlation is often preferred in other analysis, with values of 0.639, 24 0.5, 25 0.84, and 26 reported across studies, reflecting differences in ranking consistency. AUCPRC was selected as the primary evaluation metric as it provided a rigorous evaluation that focuses on minority class performance, which is critical for off-target prediction, and the abbreviated training code, as well as this assessment, are shown in Appendix A.

### 2.7. ECFPs

Figure 7 illustrates the changes in extended connectivity fingerprints (ECFPs) resulting from the progressive replacement of single atoms in a guanosine monomer structure. Starting from an all-carbon base structure, each atom was sequentially replaced with the correct atom, beginning with 1-P, the central phosphorus in the backbone. For example, line 0 represents the fingerprints of the initial all-carbon structure, line 1 shows the results after replacing one carbon with phosphorus (1-P), and line 23 corresponds to the final structure with all substitutions completed.

The fingerprints were generated using the RDKit library, where each bit in the ECFP encodes local molecular substructures. The y-axis represents atomic indices, labeled alongside their corresponding atom types, while the x-axis corresponds to fingerprint bit positions. Carbon atoms were excluded from visualization since substitutions involving carbon do not alter the fingerprints. The color intensity in the heatmap indicates the fingerprint value at each bit index. ECFPs are generated by iteratively hashing atomic environments into unique numerical identifiers. Each atom is assigned an initial feature based on its properties (e.g., atomic number, hybridization, formal charge, and bond order), with neighboring atoms being progressively incorporated up to a defined radius (radius = 1 in this study). These hashed substructures are folded into a fixed-length binary vector (256 bits), encoding the presence or absence of specific patterns. To enhance storage efficiency and facilitate visualization, the binary fingerprints were compressed into unsigned 8-bit integers using numpy’s np.packbits function, preserving the substructural information in a compact form.

Significant changes in the fingerprints were observed upon the insertion of functional group; for example, 12-O completes the sugar ring moiety, and the addition of the final nitrogen 22-N completes the nitrogenous base, so these show many signature changes. This fingerprinting method provides a powerful way to describe chemical modifications through capturing the molecular changes across the structure. However, this has limitations, such as increased memory usage due to the high dimensionality of the features. Despite its complexity, the method offers valuable insights into the impact of atomic substitutions on molecular properties.

In addition to the scores obtained using dataset 3, notable AUPRC scores were also obtained for both versions of dataset 7. These data were obtained via a process that involved structural prediction and molecular modeling. The unmodified guide-target siRNA/argonaute structure was predicted using Chai Discovery, and then the modification was inserted into the predicted structure using the experimental literature coordinates. Gromacs was then used to set up a water box with ions, and then the complex was minimized up to three times to ensure reproducible results. After minimization, distances were calculated as follows: for each minimized structure, the solvent and ions were removed, and the siRNA-hAgo2 complex was then aligned with the experimental 4f3t structure. Then, for each protein residue, the change in distance was calculated and saved as a feature. 7R provides a couple of additional data refinement steps compared to 7N: for each feature, the data were rescaled in the range of 0 to 255 based on the lowest to highest value of the column. The AUPRC scores, reaching nearly 0.75, demonstrate the utility of this method.

Our findings align with the broader trend in computational biology toward creating more comprehensive, data-driven frameworks. This progress is exemplified by powerful new AI models that are redefining both predictive and generative tasks, such as Google DeepMind’s AlphaGenome [27] and Chai-2 [28]. While AlphaGenome provides unprecedented insight into the endogenous regulatory landscape from DNA sequences and Chai-2 enables the de novo design of novel protein binders, our work addresses the complementary and equally critical challenge of predicting the off-target effects for a specific class of therapeutics—siRNAs—based on their unique structural features. A powerful future direction involves integrating these distinct capabilities. For instance, the genomic context predictions from AlphaGenome could serve as advanced input features for our structural model, while the safety predictions from our framework would be essential for evaluating candidates proposed by generative models. This synthesis of generative design, regulatory prediction, and safety assessment represents the next frontier in accelerating the development of safer, more effective therapeutics.

## 3. Materials and Methods

### 3.1. Dataset and Feature Generation

The full experimental RNA-Seq dataset is provided in the Appendix A, and was collected as part of another study [29]. This dataset captures a comprehensive array of off-target effects. From this dataset, the top 20% of genes with the highest mean expression counts were selected. Off-target effects were quantified using log_2_ fold change (log_2_FC) values derived from RNA-Seq, calculated as log_2_(treated/control). These values were then assigned as the ’labels’ in the ML algorithm, which are the target values that were being predicted. Additionally, to simplify the data, it can be effective to use only a 0 or a 1 to indicate a condition, e.g., whether the gene is off-target or not. To accomplish this, the labels were binarized via assigning a positive class (label = 1) to samples whose absolute target value exceeded the threshold of 0.25 and a negative class (label = 0) to those whose absolute target value was less than or equal to 0.25 log_2_FC.

Guide siRNA sequences were aligned to each gene to identify the intended targets. Using these targets, OpenAI’s Chai Discovery platform [30] was employed to generate 10,000 structural predictions, forming the basis for chemical modifications. These included a modified dimer at position 3 and a monomer at position 7 to represent unmodified and chemically modified siRNA strands. Molecular dynamics simulations and structural modeling were performed to refine these structures through energy minimizations and trajectory calculations, ultimately creating 30,000 unique modeled structures.

Once the structures were modeled, the key challenge became converting the structural data into numerical ’features’. For the purposes of this study, a feature is any type of data which represent the structures. We developed and applied several novel approaches, described in detail below. Our simplest and most innovative method involved computationally aligning each modeled structure to a reference experimental structure, followed by calculating the positional displacement of each residue relative to its reference. The compiled list of positional displacements was then used to derive the input features for the ML algorithm. The features derived from these structures were used to evaluate nine different modeling approaches, detailed in the subsequent sections. The initial training in TensorFlow produced mixed results; however, further optimization using Autogluon [23], under several conditions, yielded improved metrics. Notably, the multi-model ensemble approach demonstrated the best performance for datasets without structural modeling in Sets 2, 3, and 4 (Figure 1).

### 3.2. Feature Description

Hybrid datasets combining diverse feature types, such as sequence-derived and contextual features, have demonstrated improved predictive capacity in biological modeling, underscoring the importance of integrating complementary data sources into feature extraction workflows. Building on this concept, we designed nine unique datasets, each representing chemically modified siRNAs, changes in hAgo2 during RISC activation, and the targeted genes (Table 1). Eight of the nine datasets incorporate novel approaches, with Dataset 2 serving as a control. This strategy combines the strengths of molecular dynamics and structural refinement with the interpretability of conventional encodings. By adding detailed spatial and chemical insights to the feature space, this method marks a significant advancement in feature engineering for siRNA and provides a robust framework for off-target effect prediction.

The key features generated in this process included fingerprints for chemical modifications, changes in residue-to-residue distances, residue coordinates, and gene indices to help group the data. Importantly, both the fingerprints and modeling-based features are versatile and can be easily redesigned to include any modification, such as 2′-F or 2′-OMe. The total number of features varied broadly, from 43 for a simple encoding to over 2600 when extracting full xyz coordinates for each protein and RNA residue. To maintain consistency and reproducibility, the post-modeling structures were aligned with a reference, which was either the experimental structure, 4f3t [24], or the base structure obtained from Chai Discovery. Since 4f3t lacks many RNA residues, the distances for siRNAs were not calculated in datasets 0, 7N, and 7R. By using coordinates or the base structure, RNA data were obtained for datasets 1, 6N, and 6R. Recognizing the challenges regarding the trajectory’s reproducibility, we introduced minimized-only datasets (6 and 7) and added rescaling to improve feature consistency.

### 3.3. Reproducibility Assessment

Reproducibility is critical in computational structural biology. To quantitatively assess the consistency and stochastic variability of our modeling pipeline, we performed replicate runs (n = 10) for a representative subset of 1000 initial structures at key stages: (1) initial prediction via Chai Discovery, (2) post-energy minimization, and (3) post-molecular dynamics (MD) trajectory generation. For each replicate set originating from the same starting point, structures were aligned with the experimental reference 4f3t [24]. Inter-residue Cα-Cα distances within the protein were calculated. The sum of the relative standard deviations (ΣRSD) across all protein residues between replicates served as a quantitative metric of structural consistency. A lower ΣRSD indicates higher reproducibility. The calculated mean ΣRSD values were 3.39 for initial predictions, 0.03 for post-minimization structures, and, initially, 6.27 for post-trajectory structures. The exceptionally low ΣRSD after minimization highlights the convergence that was achieved. The initial trajectory ΣRSD indicated potential divergence, prompting a refinement of MD simulation parameters (specifically, a reduction in the time step; see below), which successfully reduced the mean post-trajectory ΣRSD to 3.01. This level was deemed acceptable, reflecting the consistent sampling of conformational space relevant to the derived features.

### 3.4. Minimizations and Trajectories

For the amide-modified backbone, the partial charges and initial structure were generated as previously described [31]. The energy minimizations employed the steepest descent algorithm implemented in GROMACS, run for a maximum of 50,000 steps or until a maximum force convergence criterion of 100 kJ mol^−1^ nm^−1^ was met. For the glycol nucleic acid, the initial coordinates were taken from the published structure 5v2h [21] and partial charges were derived using the restrained electrostatic potential approach based on an electrostatic potential grid calculated using the GNAU monomer at HF/6-31G* using NWChem. To incorporate the glycol nucleic acid (GNA), which lacks a conventional sugar moiety and thus precludes the fitting approach used for amide modifications, an alternative strategy was employed. The GNA base, along with its two adjacent nucleotides, was derived from the coordinates of the experimental structure. These adjacent bases served as anchor points for fitting to corresponding bases in the structure of the predicted siRNA–Ago2 complex. GROMACS [32] was used to generate a cubic solvation box ensuring a minimum distance of 1.0 nm between the solute and box edges.

MD simulations were performed using the leapfrog stochastic dynamics integrator with a total time of 0.9 ps and 10 fs per step (reduced from an initial 2 ps to improve the trajectory’s stability; see Reproducibility Assessment). Temperature was maintained at 300 K using a velocity-rescaling thermostat ($\tau_{t}$ = 0.1 ps), and pressure was maintained at 1 bar using a Parrinello-Rahman barostat ($\tau_{p}$ = 2.0 ps) with isotropic coupling. Long-range electrostatics were treated using Particle Mesh Ewald (PME) with a real-space cutoff of 1.0 nm. Van der Waals interactions used a cutoff of 1.0 nm. Protein–nucleic acid complexes were surrounded with TIP3P water molecules and ions were added to neutralize the system. Molecular dynamics and minimizations were carried out using AMBER 99 (ff99SB).

### 3.5. Chai Discovery

The predicted structures of siRNA duplexes within the hAgo2 protein were generated using Chai-Lab v0.0.1 with parameters: num_trunk_recycles = 3, num_diffn_timesteps = 200, seed = 42, and use_esm_embeddings = True. The highest-scoring prediction was retained.

### 3.6. Conversion of Ensemble Gene Identifiers to 24-bit Unsigned Integer Representation

Ensembl gene identifiers were compressed into a 24-bit unsigned integer representation to enable efficient storage and processing. Each identifier was parsed to remove the prefix ENSG and leading zeros, leaving only the numeric portion. The numeric portion was split into three unsigned 8-bit integers (uint8) corresponding to the most significant byte, middle byte, and least significant byte using bitwise operations. Specifically, the most significant byte was extracted as (n≫16)&0xFF, the middle byte as (n≫8)&0xFF, and the least significant byte as n&0xFF, where *n* is the numeric portion of the identifier. These uint8 values were stored as separate columns in the dataset, labeled gene_idx0, gene_idx1, and gene_idx2, facilitating downstream analyses. This approach ensures the preservation of uniqueness in the compressed format while minimizing storage requirements.

### 3.7. ECFPs

For datasets 0 and 4, fingerprints were generated for dimers at positions 3–4; then, monomers were generated at position 7. The fingerprints of the dimers were created through the selection of unmodified AG or amide-modified AG, using parameters nBits = 512, radius = 2. Fingerprints of monomers were obtained from either unmodified U or modified GNA U at position 7, with parameters nBits = 256, radius = 1. Binary vectors were compressed into unsigned 8-bit integers using numpy’s np.packbits, generating 64 and 32 features, respectively.

Dataset 3 utilized a positionally resolved ECFP strategy to capture detailed chemical information across both the siRNA guide strand and the intended mRNA target region. Hydrogens were first removed. Each of the 21 nucleotide positions in the siRNA guide was treated as an independent monomer. ECFPs were generated for each monomer using RDKit (nBits = 32, radius = 1), considering only atoms assigned to that specific position (via Amber topology). This yielded 21×32=672 features encoding the siRNA. Analogous positional ECFPs were generated for the relevant region of the target mRNA sequence, contributing the remaining features. The combined feature vector for Dataset 3 comprised 1344 features, providing a granular chemical description of the siRNA–mRNA interaction interface.

### 3.8. Autogluon

Hyperparameter optimization was conducted using AutoGluon’s hyperparameter_tune_kwargs function with varied trial numbers. This process targeted key models, including LightGBM, XGBoost, CatBoost, and KNN. Hyperparameter ranges were defined according to each model’s key parameters and tested across multiple predefined configurations. Trial numbers were systematically varied, with the results indicating diminishing returns after two trials. Metric optimization utilized default ensembles across metrics, such as balanced_accuracy, roc_auc_ovo_macro, recall, and f1_weighted, with ten different seed splits per dataset. Stack level optimization employed roc_auc_ovo_macro, testing num_stack_levels and time_limit across a broad range. Performance evaluation was performed using precision–recall curves.

For all training procedures, held-out validation sets were utilized. Initially, a 90/10 split was implemented for model scoring and hyperparameter optimization, after which a 95/5 split was applied. Validation sets were defined either using Stratified K-Fold (SKF) splits or automatically generated using AutoGluon’s internal methods. The full methodological details are provided in Appendix A.

## 4. Conclusions

In this work, we systematically evaluated nine distinct feature engineering strategies for predicting the off-target interactions of chemically modified siRNAs, leveraging approximately 30,000 siRNA–gene interaction pairs. Our findings establish that incorporating detailed chemical and structural information significantly enhances predictive accuracy, with positionally resolved ECFPs capturing granular chemical environments (Dataset 3, AUPRC = 0.78) and energy-minimized structural models providing crucial spatial context (Datasets 7N/7R, AUPRC = 0.74). These results highlight the complementary strengths of rich chemical descriptors and stable 3D structural representations, suggesting that future models integrating both feature types offer the greatest potential for improved performance and mechanistic insight.

The greatest advantage of this approach lies in its versatility, being able to encode any chemical modification at any position, including both novel and established modifications such as the clinically relevant 2′-F or 2′-OMe. While the predictions in this study were validated internally within our dataset to demonstrate the initial framework, extending the validation to external datasets and novel modifications is an important next step that will require additional data. Key future directions include expanding the structural context through modeling larger mRNA target regions within the hAgo2 complex using platforms such as Chai Discovery—enabling the investigation of long-range complementarity and flanking sequence effects—and refining features specifically at critical interfaces, such as the modification-proximal hAgo2 residues that are highlighted (Figure 2), to guide structure-informed experimental validation. Overall, this study provides a robust, reproducible foundation for developing structure-aware machine learning frameworks aimed at accelerating the design of safer and more effective siRNA therapeutics through improved prediction and the mitigation of off-target effects.

## Figures and Tables

**Figure 1 ijms-26-06795-f001:**
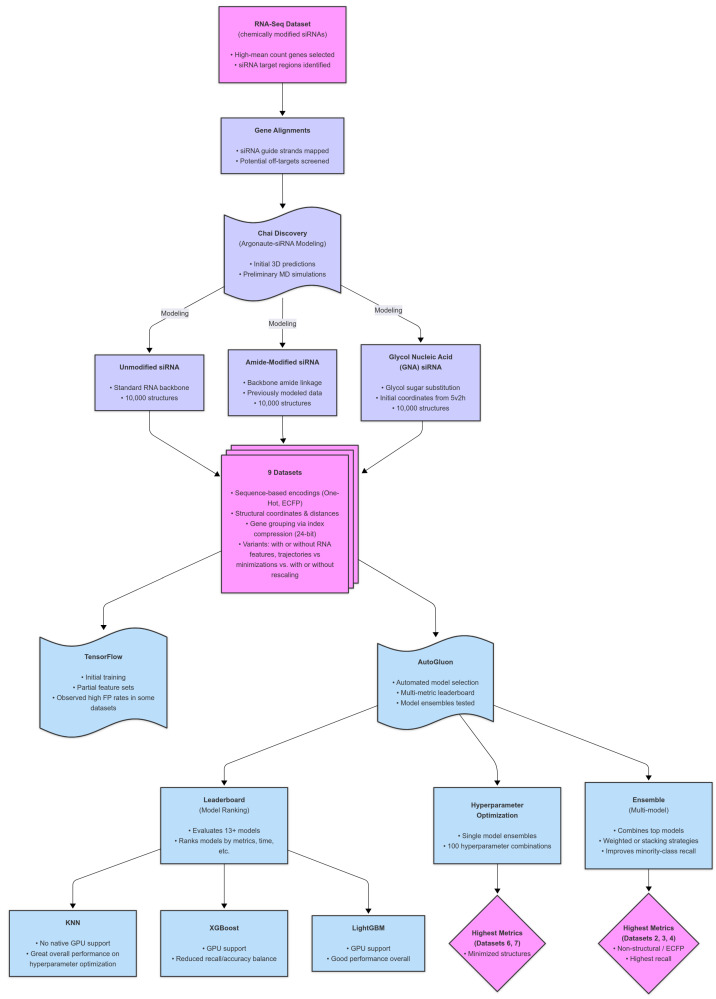
Workflow of the experimental setup. The workflow consisted of three phases: experimental, bioinformatics, and machine learning. The experimental phase included chemically modified siRNA production and RNA-Seq data collection. The bioinformatics phase involved aligning results, structural predictions, molecular modeling, and dynamics, generating nine distinct datasets. Machine learning involved initial TensorFlow tests, followed by multiple optimization rounds using AutoGluon.

**Figure 2 ijms-26-06795-f002:**
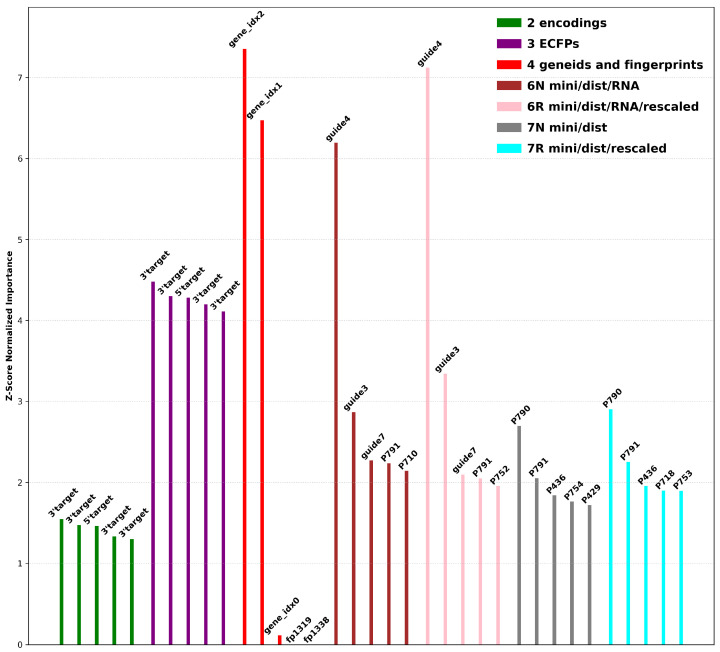
Top five features per dataset. Feature importance was assessed using RandomForestClassifier from the scikit-learn library with $n_{\mathrm{estimators}}=75$. The model was fit and the feature importance was ranked using mean decrease in impurity (Z-score normalized).

**Figure 3 ijms-26-06795-f003:**
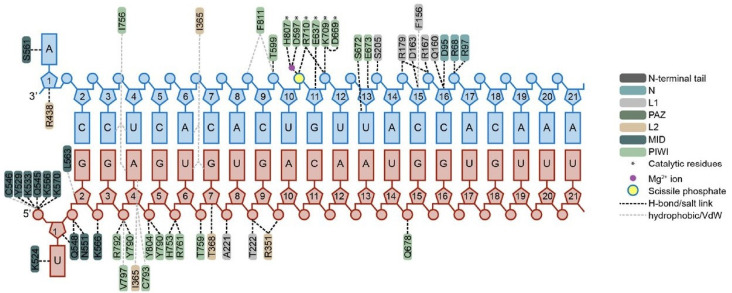
Interaction map of siRNA with hAgo2. Structural view of critical residues and domains interacting with the guide strand (red, 5′→3′) and target mRNA (blue, 3′→5′). Our analysis shows that the seed-interacting residues in the PIWI domain are the most critical features for off-target prediction. Reproduced with permission from Sarkar and colleagues.

**Figure 4 ijms-26-06795-f004:**
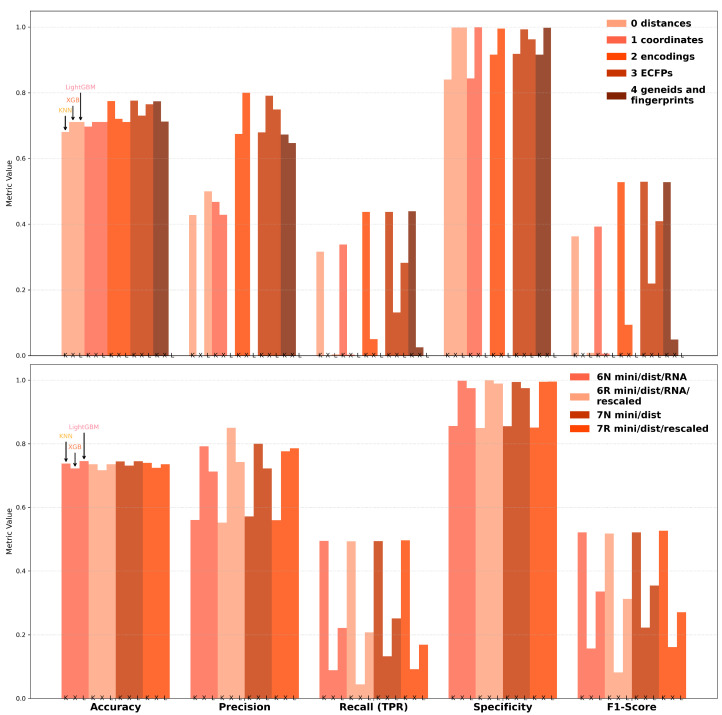
Best scoring metric by model and dataset from Hyperparameter Optimization. The results show the performance achieved for each metric after HPO, aggregated or selected from the 10-fold cross-validation process described in Appendix A. Models: KNN (K), XGBoost (X), LightGBM (L). Performance, compared considering accuracy, precision, recall (TPR), specificity, and F1-score. Missing bars indicate scores of zero, which typically occurred when a model failed to predict any true positives or true negatives for a specific fold or dataset.

**Figure 5 ijms-26-06795-f005:**
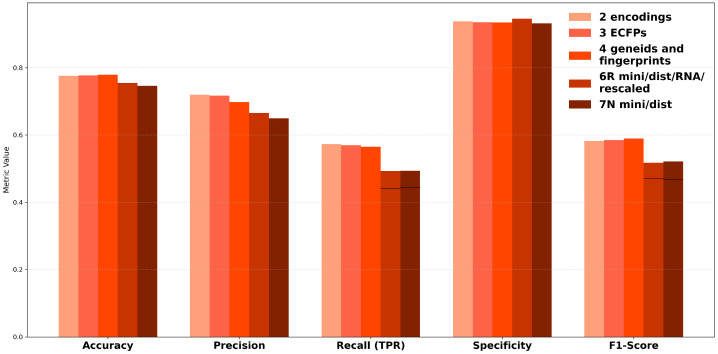
Metric optimization for top five datasets. Multi-model ensembles tested across 25 metrics to optimize minority class predictions. Selected datasets included simple encodings, chemical modification fingerprints, hybrid approaches, and structural modeling results.

**Figure 6 ijms-26-06795-f006:**
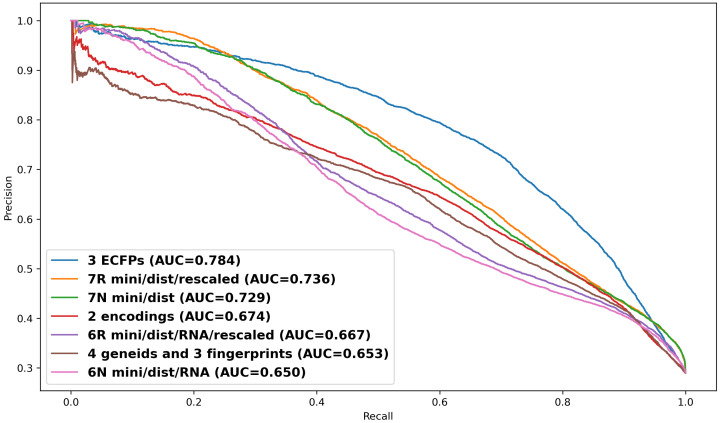
Stack level and time optimization. Dataset 3 (ECFP fingerprints) achieved the highest AUPRC (0.784), whereas both 7R and 7N (structurally modeled datasets) performed similarly (∼0.73).

**Figure 7 ijms-26-06795-f007:**
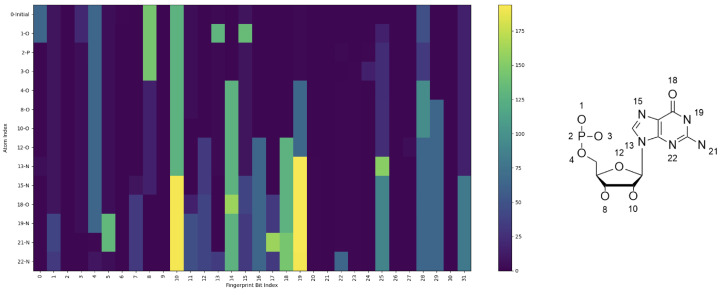
Theoretical ECFP feature demonstration (RDKit). Heatmap of ECFP changes in a guanosine monomer during sequential atomic substitutions (line 0: all-carbon; line 23: fully substituted). Y-axis lists atomic indices and types; X-axis shows fingerprint bit positions, where color intensity represents the presence of a substructure.

**Table 1 ijms-26-06795-t001:** Summary of the nine datasets used in this study.

Dataset	Description	Key Features	Features	Ref.
0	Distances calculated after modeling siRNA structures with chemical modifications.	Two molecular fingerprints containing dinucleotide at positions 3–4 and a monomer at position 7; distances calculated after trajectories.	896	4f3t
1	After modeling, full xyz coordinates of each residue stored as features.	Coordinates of all protein and RNA residues; no fingerprints; trajectories.	2637	4f3t
2	Simple RNA encoding of guide strand and predicted target with numerical modification indicators.	Encodings: A = 1, C = 2, G = 3, U = 4, dimer = 5–6, monomer = 7.	42	None
3	Extended Connectivity Fingerprints (ECFPs) from guide strand and target sequences from alignments.	As Dataset 2, using fingerprints rather than simple numerical encoding.	1344	None
4	Gene index data used as features; same fingerprints as Dataset 0.	Ensembl IDs as uint8 (gene index 0–2); fingerprints for positions 3–4, 7.	99	None
6N	Distances from minimized structures; pre-modeled structure used as reference.	Distances from pre-modeled base structure; RNA distances included.	879	Base
6R	As 6N but rescaled (0–255 range normalization).	Rescaled distances for comparability.	842	Base
7N	Distances from minimized structures; literature reference used.	Distances from the literature reference; no RNA distances.	837	4f3t
7R	Same as 7N but rescaled.	Rescaled distances for comparability.	800	4f3t

**Table 2 ijms-26-06795-t002:** Initial metric scores using AutoGluon across different datasets and models. Note: these scores reflect performance using AutoGluon’s default configurations or best models from its initial leaderboard, prior to dedicated metric, hyperparameter, or stack level optimization phases.

Dataset	Model	FN	TN	TP	FP	Precision	Accuracy (%)	TP/FP Ratio
0	LightGBM	0	0	0	0	0.00	71	N/A
1	LightGBM	46	46	72	46	0.61	72	1.57
2	LightGBM	268	308	497	267	0.65	78	1.86
4	CatBoost	243	274	471	241	0.66	78	1.95
4	KNeighbors	232	238	430	229	0.65	75	1.88
4	XGBoost	207	223	402	204	0.66	77	1.97
6N	LightGBM	194	206	304	191	0.61	74	1.59
6R	LightGBM	173	180	351	175	0.67	76	2.01
7N	LightGBM	194	207	361	193	0.65	76	1.86
7R	LightGBM	200	216	355	196	0.64	76	1.81

**Table 3 ijms-26-06795-t003:** Gene clustering via 24-bit identifier encoding. Gene identifiers (Ensembl format) were split into three 8-bit unsigned integers (gene_idx0, gene_idx1, gene_idx2) to cluster the genes into related groups.

Ensembl ID	gene_idx0	gene_idx1	gene_idx2
ENSG000000000003	0	0	3
ENSG000000000419	0	1	163
ENSG000000000457	0	1	201
ENSG000000000460	0	1	204
ENSG000000000971	0	3	203
ENSG000000001036	0	4	12
ENSG000000001084	0	4	60

## Data Availability

All datasets (including raw RNA-Seq), and the full code for feature generation, model training and eval are publicly available at https://github.com/mrichter0/siRNA-Features, accessed on 3 April 2025.

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
