# Peer review of "siRNA Features—Automated Machine Learning of 3D Molecular Fingerprints and Structures for Therapeutic Off-Target Data"

_ijms, 2025, doi:10.3390/ijms26146795_

Round 1

Reviewer 1 Report

Comments and Suggestions for Authors

The article from Richter and Admasu on structure-based chemical features for off-target prediction is a valuable addition to the field.  As interest in the field of oligo-based therapeutics continues to grow, the ability to predict on- and off-target activities of these therapies would be extremely impactful.  The paper is well-written, in particular the introduction very nicely sets the stage for work at hand.

Unfortunately, I do not believe that this work, as it is currently presented, is appropriate for the potentially general audience of this journal.  While certainly no expert in ML and computational model generation, I found it difficult to follow, including the significance of some of the results.  For example, I was not clear on the RNA sequencing study that was performed - those experimental details were unfortunately missing.  Figure 2 identifies the top 5 features for each of the different datasets, but it wasn't clear to me what those represent?  For dataset 2 in Fig. 2, "3'-target" appears 4 times - are those the same, different?  What does "3'-target" mean?  Similarly in Figure 3B, the authors point to the protein residues that they identified as important, but don't identify those residues, their interactions with the oligo, or their significance.  

While the importance of the work is clear, I was not able to clearly assess whether it would lead to a better prediction of off-targets.  Some specific examples would be needed to demonstrate the improvements made with the models and ECFPs, and subsequent predictive power.  The authors point to some reported approaches to mitigating seed-mediated off-target effects such as an amide linkage between nucleotide positions 3-4 of the guide strand, and a GNA nucleotide at position 7 of the guide strand, and include those in their modeling efforts, but I didn't understand how the model helped to better predict in the case of these particular modifications, and whether the model could predict the reported impact of those approaches.  

In summary, I appreciate the work and the value and believe it can become suitable for publication, but believe it would require more work to make it accessible to a more general audience, or alternatively the authors could consider a more specialized journal.

Reviewer 2 Report

Comments and Suggestions for Authors

The authors designed a novel computational approach for siRNA off-target prediction. This approach integrates different features including chemical modifications and structural modeling to improve the reproductivity and practicability of the prediction, potentially facilitating therapeutic design of siRNA. I believe it is a scientifically sound work that could be published. I only have a few minor questions/suggestions:

How did the authors define ‘importance’ in the ‘feature importance analysis’? How was the ‘importance score’ calculated exactly? It seems that it is not elaborated in detail in the methods.

It is stated that the crucial residues found in this work, res 790 and 791, ‘form part of the binding pocket’. Is it possible to revise Figure 3B to be a zoomed-out view to show where these residues locate in respect to the siRNA binding pocket?

Typo: in line 353, Figure 1 should be Figure 2

Round 2

Reviewer 1 Report

Comments and Suggestions for Authors

Thank you to the authors for working to modify this manuscript.  The changes are well received and it has helped to clarify to scope and context of this work.  

The additional explanation on Figure 3 in the text is appreciated, although I would strongly recommend to adjust the figure to make it clear that these are unique features, and not the same - I wasn't able to appreciate this from the figure or updated legend itself.  As someone who focuses on figures to understand the work and presented data, I could see how others (i.e. non ML experts) will struggle here.  While potentially outside of the scope of this story, I would be curious as a general audience member for a deeper discussion on the significance of these findings.  Any hypotheses to offer?

The authors should spend some additional time in the conclusions section commenting on the limitations of their model.  I understand that this is an initial framework, and as I understand it, based on unmodified RNA, with the exception of the amide linkage and a GNA residue.  All therapeutic siRNAs are heavily chemically modified with 2'-OMe and 2'-F nucleotides - how will the inclusion of those modifications impact your model?  The authors should comment on this, and I would suggest this as a good avenue for further investigation.

Some additional suggestions:

  1. The axes and labels are very difficult to read in all of the figures, even zoomed in to full page or in a printed version.  Please adjust this to make it easier on the readers.
  2. Figure 1 is in general very difficult to read - please reconstruct this so one can appreciate the workflow that you've outlined in the manuscript.
  3. Some references are missing in the introduction - for example, cite the review article referenced on line 26.
  4. Thank you to the authors for the updated title - I agree that this better reflects the scope of the work.  As an FYI for the authors, the title was not updated in the revised version for review.

Author Response

Cover Letter (Second Round Submission)

Date: April 28, 2025

To: The Editors, International Journal of Molecular Sciences
 Subject: Submission of Revised Manuscript for Second Round Review (ID: ijms-3589795)

Dear Editors,

We are pleased to submit our revised manuscript titled "siRNA Features – Automated Machine Learning of 3D Molecular Fingerprints and Structures for Therapeutic Off‑Target Predictions" (Manuscript ID: ijms-3589795) for your continued consideration in IJMS.

We greatly appreciate the constructive feedback provided by the reviewers during the previous round. Their thoughtful comments have been instrumental in helping us further improve the manuscript's clarity, scientific rigor, and accessibility.

In this revised version, we have addressed all remaining reviewer comments, including:

  • Improving figure readability by modestly increasing font sizes where appropriate (Figures 1, 2, 3);

  • Recompiling Figure 1 to enhance visibility of the workflow diagram;

  • Added additional acknowledgement of limitations in the conclusions, additional discussion of the versatility of the model to incorporate other chemical modifications;

  • Confirming the missing review article reference was incorporated in the Introduction;

  • Updating the manuscript title in the main document, as already proposed in the prior cover letter.

We have also updated the Response Letter to detail all changes point-by-point.

Thank you for the opportunity to revise our work. We look forward to your feedback and hope the manuscript is now suitable for acceptance.

Sincerely,

Michael Richter (Corresponding Author)
Department of Chemistry
Binghamton University
Email: richter@binghamton.edu

Alem Admasu PhD
Department of Physics and Astronomy
Rutgers University
Email: asa123@rutgers.edu

Response Letter (Second Round)

Manuscript ID: ijms-3589795

Title: siRNA Features – Automated Machine Learning of 3D Molecular Fingerprints and Structures for Therapeutic Off‑Target Predictions
Authors: Michael Richter, Alem Admasu
Date: April 28, 2025

Reviewer 1 Comments

Comment 1: Authors should spend time discussing limitations of their model

Response 1: - Additional statements have been added to conclusion to further indicate the initial framework: “While predictions in this study were validated internally within our dataset to demonstrate the initial framework, extending validation to external datasets and novel modifications is an important next step that will require additional data.”

Comment 2: Discussion of outside modifications such as chemically modified with 2'-OMe and 2'-F nucleotides

Response 2: Thank you for this insightful suggestion, which highlights an important aspect previously overlooked. We have expanded our manuscript accordingly, adding statements in the abstract, at the conclusion of the introduction, and in section 2.2. A considerable advantage of our approach lies in its inherent versatility, allowing the encoding or modeling of any chemical modification at any nucleotide position. Consequently, our fingerprint and modeling-based methods can be seamlessly adapted or expanded to accommodate both novel modifications as well as established, clinically relevant modifications such as 2'-F or 2'-OMe nucleotides.

Comment 3: Axes and labels difficult to read across figures.

Response 3: Thank you for noting this concern. We apologize that the original figure quality made axes and labels difficult to read due to image truncation from Word. To address this issue, we have regenerated each figure directly from the original source code at a higher resolution (300 dpi), suitable for publication. Additionally, we have increased the font sizes and made axis labels bold where possible to enhance readability, particularly in Figures 1, 2, and 3. However, for complex figures such as Figure 4, further enlargement proved impractical without compromising important structural details. We made these changes to sufficiently address the primary concern regarding figure clarity.

Comment 4: Figure 1 is very difficult to read — consider reconstruction.

Response 4: Thank you for raising this point. As mentioned above, we have recompiled Figure 1 directly from Mermaid and subsequently edited it in Adobe Photoshop to ensure the highest resolution and clarity suitable for publication. We carefully considered alternative versions and ways of restructuring the figure for improved readability. However, the current layout accurately illustrates the challenges encountered while utilizing the AutoGluon Machine Learning Platform, and we believe preserving these details will be educational and informative to readers applying our methods. 

Comment 5: Missing review article citation in Introduction.

Response 5: Thank you for pointing this out. In the revised Introduction, we have now incorporated a citation to the review article "Machine learning for siRNA efficiency prediction: a systematic review" by Martinelli et al. (Health Sciences Review, 2024), to provide appropriate background context on machine learning approaches in siRNA research. The reference has been inserted near Line 26, as suggested.

Comment 6: Title was not updated in the manuscript.

Response 6: We apologize for this oversight. The title has now been updated throughout the manuscript and matches the version provided in our previous correspondence.

Finally, as part of the revision, we have also updated the manuscript title to better reflect the scope and contribution of the work. The new title, “siRNA Features – Automated Machine Learning of 3D Molecular Fingerprints and Structures for Therapeutic Off‑Target Predictions”, emphasizes the integration of machine learning and structural modeling techniques for off-target analysis.

We sincerely thank the editors and reviewers again for their detailed comments and constructive suggestions, which have significantly strengthened the manuscript. We hope the revised version is now suitable for publication in IJMS.

Sincerely,
Michael Richter
Alem Admasu, PhD